# Relationships between Leaf Area Index and Evapotranspiration and Crop Coefficient of Hilly Apple Orchard in the Loess Plateau

**Qiong Jia [1] and Yan-Ping Wang [1,2,*]**

[1] College of Natural Resources and Environment, Northwest A&F University, No. 3, Taicheng Road, Yangling, Xianyang 712100, China; jq872464144@163.com

[2] Key Laboratory of Plant Nutrition and the Agri-Environment in Northwest China, Ministry of Agriculture, No. 3, Taicheng Road, Yangling, Xianyang 712100, China

* Correspondence: ylwangyp@163.com

**Abstract:** Drought and water shortage are the key factors that restrict the sustainable development of the apple industry in the Chinese Loess Plateau. The accurate prediction of ET can provide a scientific basis for water management of apple orchards. A study on the relationship between LAI, ET and crop coefficient Kc under water deficit is particularly necessary for the accurate prediction of ET in apple orchards. In this work, the crop coefficient Kc under water deficit was defined as the product of the crop coefficient $Kc_I$ under no water stress and the water stress coefficient Ks, namely $Kc = Kc_I \times Ks$. LAI and ET of the hilly apple orchard were measured from April to September in 2019 and 2020. The results showed: (1) The LAI of the apple orchard showed a trend of rapid increase—moderate increase—declined during the growth period, with 0.26–2.16 [$m^2$ $m^{-2}$] variation range; (2) The ET of the orchard was greater than the rainfall, the maximum ET was in July or August. The maximum components of ET in the apple orchard was E, with 47.8–49.1% of ET; T accounted for 42.5–43.9% of ET; Ic accounted for only 9.1–9.6% of ET; (3) There was a significant exponential relationship between the LAI and T or ET. The crop coefficient $Kc_I$ under no water stress changed with the development of the apple tree canopy. The variation of water stress Ks was basically consistent with the variation of rainfall; (4) There is a significant exponential relationship between LAI and crop coefficient Kc under water deficit ($Kc = 0.1141e^{1.0665LAI}$, $R^2 = 0.7055$, $p < 0.01$). This study demonstrates that LAI could be used to estimate the crop coefficient Kc of apple orchards under water deficit in the Loess Plateau, and the actual evapotranspiration of apple orchards in this region could be predicted.

**Keywords:** leaf area index; evapotranspiration; crop coefficient; hilly apple orchard; Loess Plateau

## 1. Introduction

The Loess hilly region of northern Shaanxi is a high-quality apple (*Malus domestica*) production region due to its vast land resources, abundant sunlight, large temperature difference, good ventilation and low pollution, and the apple area has exceeded 1.3 million $hm^2$ by 2020, which increased farmers' income and reduced soil loss. However, less rainfall (usually only 450–550 mm), large soil evaporation and strong transpiration in this area result in serious soil moisture deficits and widespread deep soil drying in the hilly apple orchard. Soil water deficit has a great negative effect on apple production.

Evapotranspiration (ET) is an important component of water balance in apple orchards, which consists of soil evaporation (E), transpiration (T) and canopy interception (Ic) [1]. The main factors affecting ET are the canopy area and architecture, evaporative power of the atmosphere ($ET_0$) and stomatal conductance and soil type [2]. $ET_0$ is defined as the evapotranspiration rate at which water would be removed from a reference surface where water is not limited or a limited factor [2]; it is necessary to accurately measure/quantify it under varied data-availability conditions. Despite the necessity of taking decisions on

sectoral water distributions, irrigation scheduling, groundwater recharge, and reservoir management at both the spatial and temporal scales, in situ measurement of ET is still a difficult task [3,4]. The widely used equations, which have been developed to estimate $ET_0$, include the California Irrigation Management Information System Penman equation, the Penman–Monteith equation standardized by the Food and Agriculture Organization (FAO), the Reference Evapotranspiration equation standardized by the American Society of Civil Engineers, the Hargreaves equation and Priestley-Taylor model [5–10], besides, indirect ET estimation methods, such as moderate resolution imaging spectroradiometer (MODIS), satellite-based remote-sensing techniques and the water-budget approach built into the semidistributed variable infiltration capacity (VIC-3L) land-surface model [3]. The FAO's suggested approach of the FAO-56 Penman-Monteith model (FAO-$ET_0$) is recommended as the standard model world-wide for estimating $ET_0$ [2,5]. The FAO-$ET_0$ require meteorological data from a well-watered, 12-cm-high grassy surface that fully covers the ground to estimate $ET_0$ [11], as the use of inappropriate data for $ET_0$ estimation from non-ideal surfaces, leads to significant and systematic cumulative errors introducing uncertainties when determining the crop water requirements in a region [12]. The major limitation of the FAO-$ET_0$ is the high requirement of several meteorological variables that are often incomplete and/or unavailable [13,14]. The application of the FAO-$ET_0$ equation has been reported to lead to errors, especially when estimated variables as an alternative to measured values were used as inputs in $ET_0$ computations [15,16]. The most important meteorological variables for $ET_0$ estimation are air temperature and solar radiation [17,18], and daily temperature range could be related to relative humidity and cloudiness [19]. In spite of advection, which depends on the interaction between temperature, relative humidity, vapor pressure, and wind speed [20], the air temperature is the most widespread monitored meteorological variable among the required inputs for $ET_0$ estimation [21].

Measuring crop evapotranspiration ($ET_c$) and relating it to reference evapotranspiration ($ET_0$) is the standard procedure for the determination of crop coefficient ($K_c$) used for skilled irrigation management [11,22]. Kc is defined as the ratio between the actual crop evapotranspiration ($ET_c$) and reference evapotranspiration ($ET_0$), i.e., $K_c = ET_c/ET_0$. Since biotic and abiotic stress on the crop may affect its water consumption [2,23], standard $K_c$ needs to be determined on plants that are disease-free, well-fertilized and achieving full production and grown in a large field under optimum soil water conditions [11]. $K_c$ varies along the growing season as a function of leaf area index (LAI) dynamics, the solar radiation intercepted by the canopy and the phenological stage of the crop [11,22]. The $K_c$ values may vary with agricultural practices [24]. $K_c$ has two components–$K_e$, soil evaporation, and $K_t$, plant transpiration, i.e., $Kc = K_e + K_t$ [11].

At present, $ET_c$ in a cultivated horticultural field has been measured and estimated with different techniques, such as the microclimatological method [25], soil water balance method [26] and the method of combining measurements of soil evaporation with plant transpiration [27–29]. In addition, remote sensing [30,31] and lysimeter [32–35] techniques have also been applied to the determination of $ET_c$. Lysimetry is considered the standard technique for measuring $ET_c$. The soil water balance is a simple method for estimating $ET_c$; however, it requires accurate estimates of its components, such as deep drainage and change in soil water content. There are difficulties in using the micrometeorological method because of canopy heterogeneity of orchards [36]. As well, remote sensing and lysimeter techniques have high costs of equipment. The combination of soil evaporation and plant transpiration measurements was proven to be more accurate than the microclimatological method and soil water balance method [37].

LAI is closely related to photosynthesis, transpiration, evapotranspiration and productivity [38]. Vegetation has a major impact on the different hydrological components of the water cycle which alters the surface energy balance. Heterogeneity in vegetation affects the physical characteristics, such as albedo, atmospheric transmissivity, root zone soil moisture, crop growth, evapotranspiration, leaf area index (LAI), stomatal conductance and surface runoff [39–41]. Crop coefficient Kc reflects the comprehensive influence of crop growth

conditions, biological characteristics, yield level and other conditions on water demand. Crop water demand can be calculated according to reference evapotranspiration and crop coefficient Kc [42]. Crop coefficient Kc is mainly affected by plant growth and development with the change of the growing season, and is related to leaf area index (LAI) dynamics, canopy intercepted solar radiation and crop phenology [43].

Since water stress can affect crop growth and yield, it is important to estimate evapotranspiration accurately, especially in semi-arid regions [44]. Munitz, etc. [43] reported that the canopy area of wine grapevines is a reliable approach for estimating their Kc in semi-arid and arid regions with a limited water supply. Wang, etc. [45] developed appropriate region-specific crop coefficients (Kc, Ks), and both Kc and Ks to LAI is a linear regression, but to canopy conductance is an exponential one. LAI is a better indicator than canopy conductance when it is used to predict Kc and Ks of vineyards in arid Northwest China. Du, etc. [46] reported that a significant correlation was detected between LAI and transpiration at the stages of leaf expansion and fruit expanding; however, there was no correlation between LAI and transpiration at the stages of bud development and flowering, and fruit maturing. Average LAI across the whole apple growth season was significantly higher in the treatments with irrigation amounts of 500 mm than 400 mm in arid Northwest China. Gong, etc. [47] reported that the crop coefficient Kc showed a strong linear dependence on the leaf area index of apple trees. The water stress coefficient Ks was approximately 1.0 when soil moisture was above 23% and started to decrease linearly after that. The prediction of evapotranspiration in apple orchards can be made using the Food and Agriculture Organization's crop coefficient method from commonly available meteorological data in arid Northwest China.

Crop coefficient Kc was greatly affected by climate, soil, water stress degree and other environmental conditions in different regions [48]. Significant uncertainties in estimating ET and its components have been reported in previous studies when directly using the FAO-proposed coefficient values in different regions [49]. Thus, it is necessary to identify the specific values of Kc and Ks across different agricultural ecosystems and environmental conditions, as these values provide important guidance for local irrigation practices and can be used to improve water-use efficiency [50]. The relationships of both Kc and Ks to various ecological and environmental factors (e.g., canopy conductance, LAI, and water deficit) need to be investigated in detail for the accurate prediction of evapotranspiration of apple orchards on Chinese Loess Plateau. However, as far as we know, such studies are relatively few.

In this study, the crop coefficient Kc was defined as the product of the crop coefficient $Kc_I$ under no water stress and the water stress coefficient Ks under the condition of water stress, namely $Kc = Kc_I \times Ks$. We selected nine apple trees 5–9-years-old in Mizhi County to measure the LAI and evapotranspiration of apple orchards in hilly apple orchards for two consecutive years. The objectives of this study were to: (1) Identify the dynamic variation characteristics of LAI in hilly apple orchards; (2) Analyze the change of ET of apple orchards and quantify the components of ET; (3) Reveal the relationship between LAI and ET; and (4) Determine the crop coefficient $Kc_I$ under no water stress and the water stress coefficient Ks at different growth stages of apple trees, and establish the relationship between LAI and Kc under water deficit. It provides scientific reference for accurate prediction of evapotranspiration and guidance for water balance management of apple orchards in Chinese Loess Plateau.

## 2. Materials and Methods

### 2.1. Study Area

The field study was conducted from April to September 2019 and 2020 at the Mizhi Dangta Mountain Apple Technology Demonstration Base (38°08′32″ N, 109°57′20″ E), located in the Yulin Mountain Apple Test Station, Shanxi Province of northwest China. Mizhi County is located in the Loess Plateau with an altitude of 847.2–1255.2 m. The site is a semi-arid climate in the temperate zone, where the annual average temperature and

total radiation are 8.9 °C and 587.1 kJ cm$^{-2}$, respectively, and the annual average rainfall is 40–500 mm, where most of the rainfall from April to June was an ineffective rainfall of less than 5 mm, and more than 60% of the rainfall occurred from July to September. Moreover, the rainfall is mostly in the form of heavy rain with high intensity. The total frost-free period was 162 days. The soil type of the experimental site is yellow cotton with an average bulk density of 1.20–1.35 g cm$^{-3}$. The field water capacity and soil organic matter content were 22.4% and 4.2 g·kg$^{-1}$, respectively.

The area of the experimental apple orchard is 13.5 hm$^2$; the main varieties are *Red Fuji*, pollinating varieties are *Gala* and *Qinguan* and so on. The plant row spacing was 4 m × 5 m. Nine *Red Fuji* apple trees of different sizes were selected as sample plants; their canopy structure is shown in Table 1, of which 3 apple trees with the same age and growth status were irrigated (the number is 7, 8 and 9) in order to determine the crop coefficient Kc$_I$ under no water stress. In order to avoid water stress in the apple tree growing season, timely irrigation should be carried out when the soil water content is less than or close to 60% field capacity. Considering that the uneven leaf coverage of apple trees leads to the heterogeneity of solar radiation on the soil surface, 5 micro-lysimeters were installed around each fruit tree, and the location is shown in Figure 1. A field weather station was installed in the center of the orchard to monitor meteorological data. Management measures, such as fertilization, weed and insect control were the same for all sample plants.

**Table 1.** The canopy structure of apple trees.

| Tree Number | Tree Age (Years) | Canopy Diameter (cm) | Canopy Height (cm) | Stem Height (cm) | Diameter (cm) | Canopy Volume (m$^3$) | Land Area (m$^2$) |
|---|---|---|---|---|---|---|---|
| 1 | 7 | 401.0 | 340.0 | 75.0 | 10.8 | 28.6 | 12.6 |
| 2 | 5 | 317.5 | 320.0 | 56.0 | 6.7 | 16.9 | 7.9 |
| 3 | 7 | 442.5 | 310.0 | 73.0 | 12.4 | 31.8 | 15.4 |
| 4 | 6 | 315.0 | 220.0 | 79.0 | 7.3 | 11.4 | 7.8 |
| 5 | 9 | 458.5 | 216.0 | 90.0 | 11.2 | 23.8 | 16.5 |
| 6 | 7 | 399.5 | 231.0 | 81.0 | 13.0 | 19.3 | 12.5 |
| 7 | 7 | 401.5 | 270.0 | 77.0 | 12.2 | 26.6 | 14.3 |
| 8 | 7 | 413.5 | 282.0 | 83.0 | 14.1 | 28.7 | 16.2 |
| 9 | 7 | 420.1 | 295.0 | 82.0 | 13.8 | 29.4 | 17.1 |

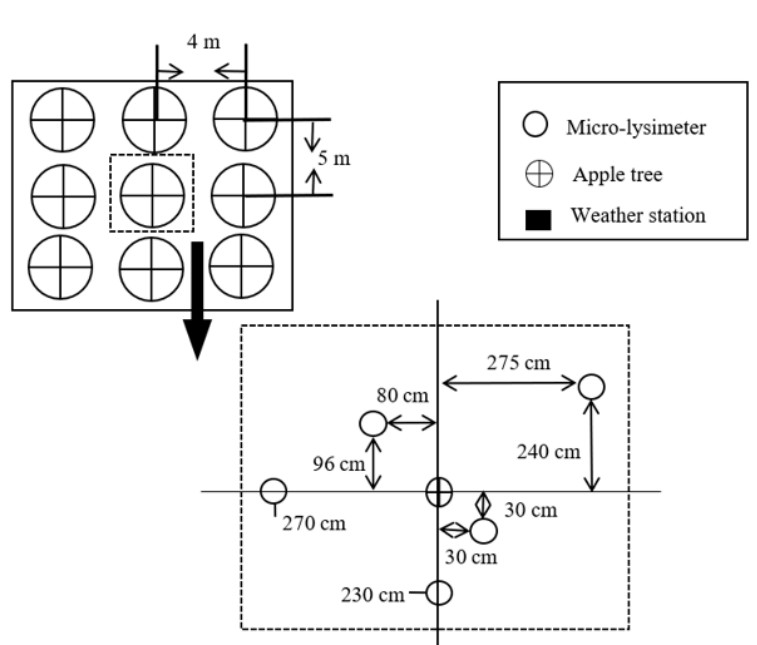

**Figure 1.** Location of micro-lysimeters.

## 2.2. Leaf Area Index Measurement

The leaf area index (LAI) was measured using a standard digital SLR camera equipped with an optical fish-eye lens and was used to take 8 pictures of each apple tree [51]. LAI was measured every 15 days during the apple growth period from April to September in 2019 and 2020. In order to make sure the camera was level and shoot up or down depending on the height of the plant, and to ensure the accuracy of the measurements, canopy photographs were taken before sunrise or after sunset on cloudy and sunny days. The captured images were processed using CAN_EYE Software V 6.47 and the average LAI was obtained using CAN_EYE V 5.1.

## 2.3. Monitoring Soil Water Contents

Soil samples were collected at intervals of 20 cm in a 0–200 cm soil profile using a soil auger. Soil moisture was measured at 5–7 day intervals by oven at 105 °C drying method during the experiment, increased the number of times before and after irrigation and delayed by a day or two when rainfall events occurred. Soil water storage was obtained by integrating soil water content, soil bulk density and respect to depth.

## 2.4. Measurement of Evapotranspiration

The total evapotranspiration of apple trees was calculated by adding the transpiration (T), soil evaporation (E) and canopy interception (Ic). Evapotranspiration can be expressed as:

$$ET = E + T + Ic \tag{1}$$

### 2.4.1. Soil Evaporation

Soil evaporation of 0–20 cm was measured with a micro-lysimeter [52] made from PVC tubes, with 11 cm in diameter and 20 cm in height during non-rainfall periods. Five micro-lysimeters were installed and numbered around each apple tree, respectively (Figure 1), and there were 45 micro-lysimeters in total. The micro-lysimeters were weighed every day at 08:00 using an electric balance with a precision of 0.1 g. The soil in the micro-lysimeters was replaced every 3–5 days. Soil evaporation (E, mm $d^{-1}$) was calculated as follows:

$$E = 10 \times \frac{\Delta W / \rho}{\pi (D/2)^2} \tag{2}$$

where the value of 10 is the conversion factor; $\Delta W$(g) is the weight change of the micro-lysimeter in 24 h; $\rho$ is the density of water (g $cm^{-3}$); D is the internal diameter of the micro-lysimeters (cm).

### 2.4.2. Transpiration

The SF velocities (L $h^{-1}$) of selected experimental apple trees were measured routinely by using thermal-dissipation probes [53,54]. A pair of probes (model: TDP-10) with 10 mm in length and approximately 1.2 mm in diameter were inserted into the trunk sapwood of each tree (one heated and one unheated). The monitoring data were recorded every half an hour. The sap-flux velocity of individual trees was calculated as [55,56]:

$$SF = A_s \times 0.0119 \left( \frac{\Delta T_{max} - \Delta T}{\Delta T} \right)^{1.231} \times 3600 \tag{3}$$

where As is the sapwood area ($cm^2$); $\Delta T_{max}$ is the maximum temperature difference (°C) of the two probes (the heated and unheated) when there is no liquid flow before dawn; $\Delta T$ is the difference in instantaneous temperature difference (°C) between the heated and unheated probes. The results of $\Delta T_{max}$ and $\Delta T$ are obtained automatically by the instrument.

In order to avoid damaging the sampled trees and ensure the continuity of the experiment, the sapwood area was measured outside the experimental plot and the relationship

was determined between the sapwood area and diameter at breast height (DBH) for calculating daily cumulative SF [57] (Figure 2).

The daily transpiration of apple ($T_{day}$, mm) can be calculated as:

$$T_{day} = \frac{n \times SF_{day}}{S} = \frac{n \times 24 \times \overline{SF}}{S} \tag{4}$$

where n is the number of selected apple trees; $SF_{day}$ is the daily liquid flux of a single tree (L day$^{-1}$); S is the experiment plot area (m$^2$); $\overline{SF}$ is the the average sapwood liquid flux for 24 h (L h$^{-1}$).

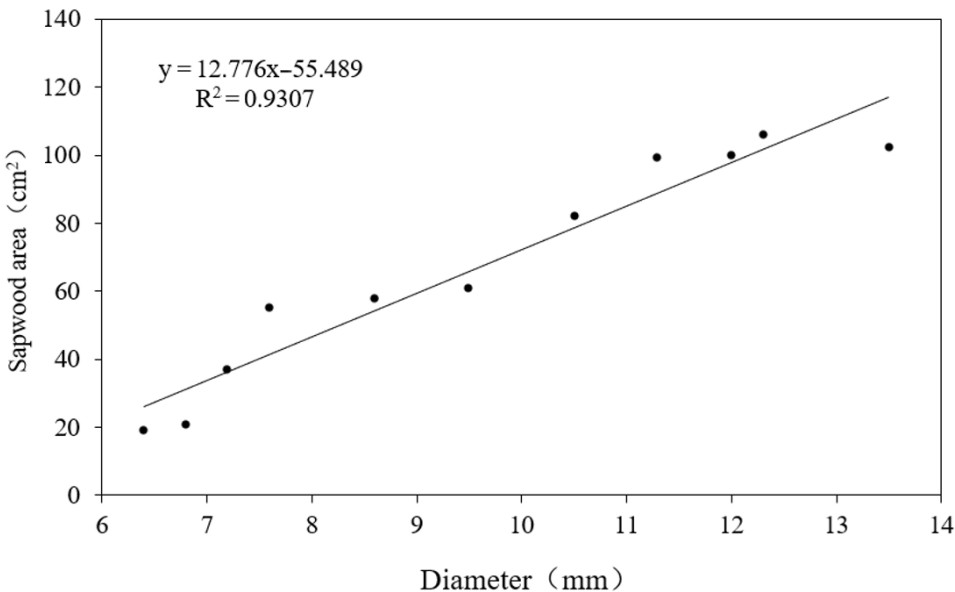

**Figure 2.** Relationship between diameter and sapwood area at the beginning of study in 2019.

### 2.4.3. Canopy Interception

The canopy interception was estimated by subtracting stemflow and throughfall from precipitation; the calculation formula is as follows:

$$Ic = P - S_tF - T_s \tag{5}$$

where P is gross precipitation (mm); $T_s$ is throughfall (mm); $S_tF$ is stemflow (mm).

Throughfall ($T_s$): During the study period, a homemade rain gauge was used to measure penetrating rainfall under the canopy of selected trees. $T_s$ was recorded within 30 min of each gross precipitation event. The recorded values were converted to water depths (mm) per unit area [58].

Stemflow ($S_tF$): Corrugated plastic tubes (3.0 cm diameter) were longitudinally halved, the contact between the lower edge of the plastic tubes and the trunk of the tree was sealed with silicone, and the end of the plastic tubes was connected to a 20 L plastic bucket. The data were recorded 30 min after the end of the rainfall event. The measured water quantity divided by the average canopy projected area of a single apple tree was converted to stemflow depth (mm) [59].

### 2.5. Meteorological

Precipitation, net radiation, air temperature, relative humidity, wind speed and soil heat flux were continuously monitored by a small farmland weather station produced by Campbell Scientific, located in the center of the experimental orchard, and the data were recorded every 30 min. In order to meet the requirements of $ET_0$ estimated by the PM equation, irrigation was carried out irregularly around the weather station. Since late April, grass with a height of more than 10 cm was always fully covered on the surface.

### 2.6. Reference Evapotranspiration

Daily reference crop evapotranspiration ($ET_0$, mm day$^{-1}$) was calculated using the widely used FAO56-Penman-Monteith equation [2]. The calculation is as follows:

$$ET_0 = \frac{0.408\Delta(R_n - G) + \gamma\frac{900}{T+273}U_2(e_s - e_a)}{\Delta + \gamma(1 + 0.34U_2)} \tag{6}$$

where $R_n$ is the net radiation (mJ m$^{-2}$ d$^{-1}$); G is soil heat flux density (MJ m$^{-2}$ d$^{-1}$); $\gamma$ is the hygrometer constant (kPa $°C^{-1}$); T is the mean air temperature at 2 m height ($°C$); $U_2$ is the wind speed at 2 m height (m s$^{-1}$); $e_s - e_a$ is the difference of saturated vapor pressure (kPa); $\Delta$ is the slope of saturated vapor pressure with temperature (kPa $°C^{-1}$).

### 2.7. Crop Coefficient Calculations

Crop coefficient $Kc_I$ under no water stress can be determined by the ratio of evapotranspiration to reference evapotranspiration under irrigation. The calculation is as follows:

$$Kc_I = \frac{ET_I}{ET_0} \tag{7}$$

where $ET_I$ is the evapotranspiration under irrigation conditions (mm d$^{-1}$); $ET_0$ is the reference evapotranspiration calculated by the Penman-Monteith equation (mm d$^{-1}$).

The water stress coefficient Ks can be calculated according to the measured evapotranspiration and reference evapotranspiration under the condition of water deficit and the crop coefficient $Kc_I$ under no water stress [2]. The calculation is as follows:

$$Ks = \frac{ET_D}{ET_0 \times Kc_I} \tag{8}$$

where $ET_D$ is the evapotranspiration under water deficit (mm d$^{-1}$); $ET_0$ is the reference evapotranspiration calculated by Penman-Monteith equation (mm d$^{-1}$); $Kc_I$ is the crop coefficient without water stress.

Crop coefficient Kc can be calculated from the crop coefficient $Kc_I$ and water stress coefficient Ks under the condition of no water stress, and the calculation is as follows:

$$Kc = Kc_I \times Ks \tag{9}$$

where $Kc_I$ is the crop coefficient without water stress; Ks is the water stress coefficient.

### 2.8. Statistical Analysis

The measured test data were statistically analyzed by Excel in WPS, plotted by Excel and R language, and analyzed by SPSS19.0 for correlation and regression.

## 3. Results

### 3.1. Leaf Area Index

The change of the LAI in the apple orchard showed a rapid increase—moderate increase—declined trend from April to September in the Loess hilly region of northern Shaanxi in China (Figure 3). The change of LAI was small at the bud development period, then from mid-April to June, the LAI increased rapidly from 0.38 [m$^2$ m$^{-2}$] to 1.67 [m$^2$ m$^{-2}$] in 2019 and from 0.26 [m$^2$ m$^{-2}$] to 1.70 [m$^2$ m$^{-2}$] in 2020, due to new branch growth and leave enlargement, and at this stage apple trees need enough water to maintain normal physiological activities [24]. From July to mid-August, the LAI slowly increased from 1.67 [m$^2$ m$^{-2}$] to 1.83 [m$^2$ m$^{-2}$] in 2019 and from 1.70 [m$^2$ m$^{-2}$] to 1.86 [m$^2$ m$^{-2}$] in 2020; this is because the new shoots stopped growing and the leaf size was stable. After mid-August, the autumn treetop of apple trees grew to produce new leaves, which gradually expanded and LAI increased slightly. By mid-September, the LAI reached its maximum

value of 2.06 [$m^2$ $m^{-2}$] in 2019 and 2.16 [$m^2$ $m^{-2}$] in 2020 for the entire growing season; after that, LAI slightly went down, which may be attributed to rainstorm, strong wind and other climatic conditions.

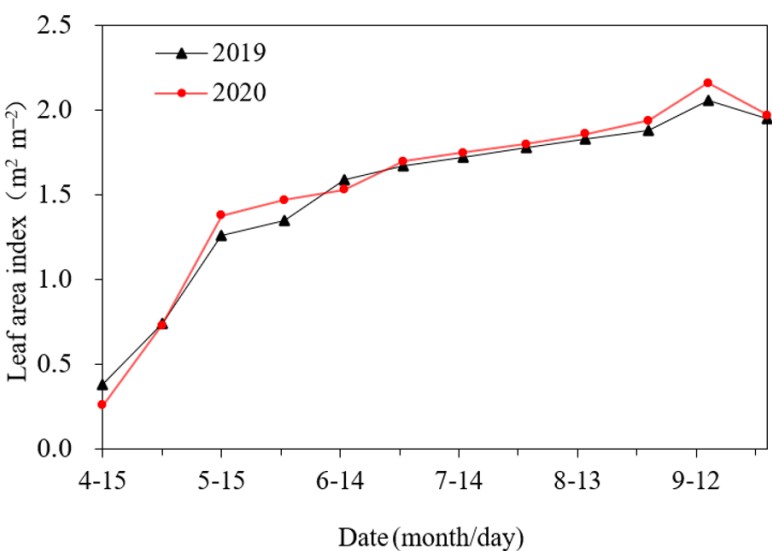

**Figure 3.** Dynamics of leaf area index during apple tree growth period in 2019 and 2020.

### 3.2. Precipitation and Reference Evapotranspiration

The rainfall distribution of the experimental orchard from April to September in 2019 and 2020 is shown in Figure 4, and 49 and 57 rainfall events were observed from April to September in 2019 and 2020, respectively, with accumulative rainfall totals of 399.5 mm and 411.9 mm, respectively. The maximum rainfall occurred on 22 July 2019 and 5 August 2020, with 53.5 mm and 95 mm, respectively. The range of monthly rainfall in 2019 and 2020 was 8.9–101.1 mm and 10.7–245.2 mm, respectively (Table 2), and the rainfall distribution was extremely uneven. The rainfall from July to September accounted for 70.8% and 90.0% of the total rainfall in the growth period (from April to September), respectively. Rainfall was the least in May and the most in July and August, with high intensity and long duration, which provided sufficient water for the soil and effectively alleviated the water stress.

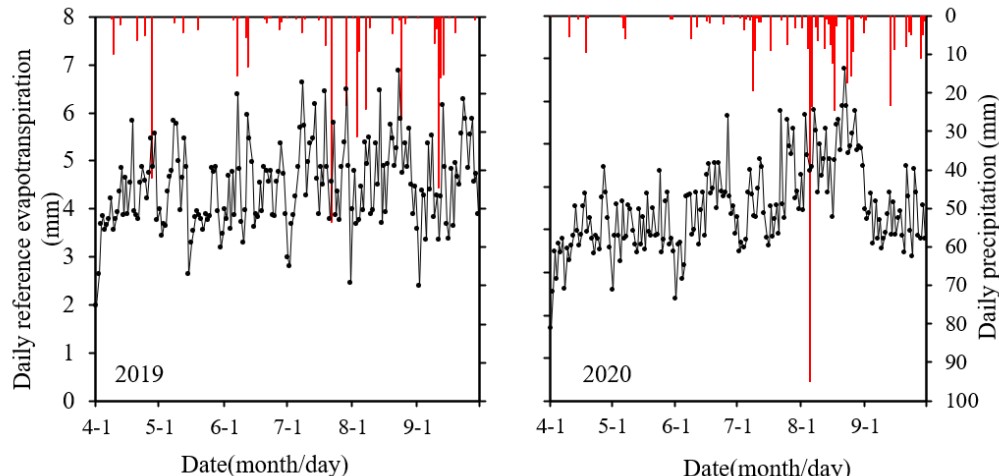

**Figure 4.** Dynamics of daily precipitation and reference evapotranspiration of apple orchards during the growth period in 2019 and 2020.

Daily $ET_0$ in 2019 and 2020 ranges from 1.98–6.88 mm $d^{-1}$ and 1.68–7.76 mm $d^{-1}$, respectively, with an average of 4.48 mm $d^{-1}$ and 4.52 mm $d^{-1}$, respectively. The maximum daily $ET_0$ occurred on 23 August 2019 and 22 August respectively. Total $ET_0$ in 2019 and

2020 was 829.9 mm and 842.6 mm, respectively (Table 2). The range of monthly $ET_0$ in 2019 and 2020 was 125.5–151.9 mm and 114.2–192.7 mm, respectively (Table 2). From April to September, the monthly $ET_0$ showed a trend of first increase and then decrease, with the maximum monthly $ET_0$ in August and the minimum monthly $ET_0$ in April.

**Table 2.** Variation of monthly precipitation (P) and monthly reference evapotranspiration ($ET_0$) of apples from April to September in 2019 and 2020.

| Year | Indice | April | May | June | July | August | September | Sum |
|------|--------|-------|-----|------|------|--------|-----------|-----|
| 2019 | P (mm) | 65.5 | 8.9 | 42.4 | 90.5 | 101.1 | 91.1 | 399.5 |
| | $ET_0$ (mm) | 125.5 | 131.9 | 133.8 | 148.1 | 151.9 | 138.7 | 829.9 |
| 2020 | P (mm) | 16.3 | 10.7 | 14.2 | 58.3 | 245.2 | 67.2 | 411.9 |
| | $ET_0$ (mm) | 114.2 | 125.0 | 135.6 | 148.8 | 192.7 | 126.3 | 842.6 |

### 3.3. Evapotranspiration

Monthly soil evaporation (E), transpiration (T), canopy interception (Ic) and evapotranspiration (ET) of the apple orchard during the study period in 2019 and 2020 are shown in Table 3. Annual ET was 424.8, 463.2 mm for 2019 and 2020, respectively. The ET in both 2019 and 2020 was greater than the respective rainfall, indicating that the orchard presents a negative water balance. Monthly ET ranged from 29.3 to 122.3 and 26.7 to 175.6 mm in 2019 and 2020, respectively. The trends of monthly ET were similar in each year. Monthly ET was highest in August for both years. The maximum components of ET in the apple orchard was soil evaporation in both 2019 and 2020, which were 203.3 mm and 227.5 mm, respectively, accounting for 47.8% and 49.1% of the total ET, respectively. The annual T were 180.6 mm and 203.2 mm for 2019 and 2020, which accounted for 42.5% and 43.9% of the annual ET, respectively. The minimum components were Ic, which were 41.0 mm and 42.3 mm for 2019 and 2020, respectively, accounting for 9.6% and 9.1% of the ET, respectively.

**Table 3.** Variation of monthly evapotranspiration of apples from April to September in 2019 and 2020.

| Year | Month | E (mm) | T (mm) | Ic (mm) | ET (mm) | Percentage (%) |
|------|-------|--------|--------|---------|---------|----------------|
| 2019 | 4 | 29.1 | 4.0 | 2.5 | 35.6 | 8.4 |
| | 5 | 18.2 | 10.3 | 0.9 | 29.3 | 6.9 |
| | 6 | 23.5 | 20.5 | 3.7 | 47.7 | 11.2 |
| | 7 | 40.2 | 49.4 | 11.2 | 100.8 | 23.7 |
| | 8 | 52.3 | 57.2 | 12.8 | 122.3 | 28.8 |
| | 9 | 40.0 | 39.2 | 9.9 | 89.0 | 21.0 |
| | Total | 203.3 | 180.6 | 41.0 | 424.8 | 100.0 |
| 2020 | 4 | 21.9 | 3.6 | 1.2 | 26.7 | 5.8 |
| | 5 | 16.4 | 11.6 | 1.0 | 29.0 | 6.3 |
| | 6 | 26.7 | 24.9 | 0.9 | 52.5 | 11.3 |
| | 7 | 40.6 | 47.6 | 7.0 | 95.2 | 20.6 |
| | 8 | 86.3 | 75.5 | 23.8 | 175.6 | 37.9 |
| | 9 | 35.7 | 40.0 | 8.4 | 84.2 | 18.2 |
| | Total | 227.5 | 203.2 | 42.3 | 463.2 | 100.0 |

Note: The percentage (%) indicates monthly ET/the total ET from April to September.

In April 2019, the apple trees entered the germination and flowering period, the branches and leaves were gradually developing, due to heavy rainfall, E was relatively intense and the ET was also larger, accounting for 8.4% of the total ET. The ET decreased in May, only accounting for 6.9%. Since June to August entered the rainy season, the rainfall increased and the temperature rose rapidly. The water consumption of the apple trees was intense, the ET gradually increased and reached the maximum value of 122.3 mm in August, accounting for 28.8% of the total ET. In September, the ET decreased to 89.0 mm. In the two years, the ET in April 2020 was the smallest, only 26.7 mm, accounting for 5.8%

of the total annual ET. The ET increased continuously from May to July in 2020, accounting for 6.3%, 11.3% and 20.6% of the total ET, respectively. In August, the rainfall was larger and the ET increased, which accounted for 37.9% of the total ET. The ET in September decreased, and only accounted for 18.2% of the total ET.

### 3.4. Leaf Area Index and Evapotranspiration Relationship

The exponential relationship between LAI and evapotranspiration (ET) or transpiration (T) was strong and significant (Figure 5, $p < 0.01$). This indicates that LAI is an important factor affecting T and ET. In 2019, the fitting equation of LAI and T was as follows: $Y = 1.0615e^{1.5494X}$ ($R^2 = 0.8206$, $n = 72$); the fitting equation of LAI and ET was $y = 9.2836e^{0.7983x}$ ($R^2 = 0.5334$, $n = 72$). In 2020, the fitting equation of LAI and T was as follows: $Y = 0.6911e^{1.6856x}$ ($R^2 = 0.8011$, $n = 72$); the fitting equation of LAI and ET was as follows: $Y = 7.7238e^{0.9009x}$ ($R^2 = 0.5417$, $n = 72$).

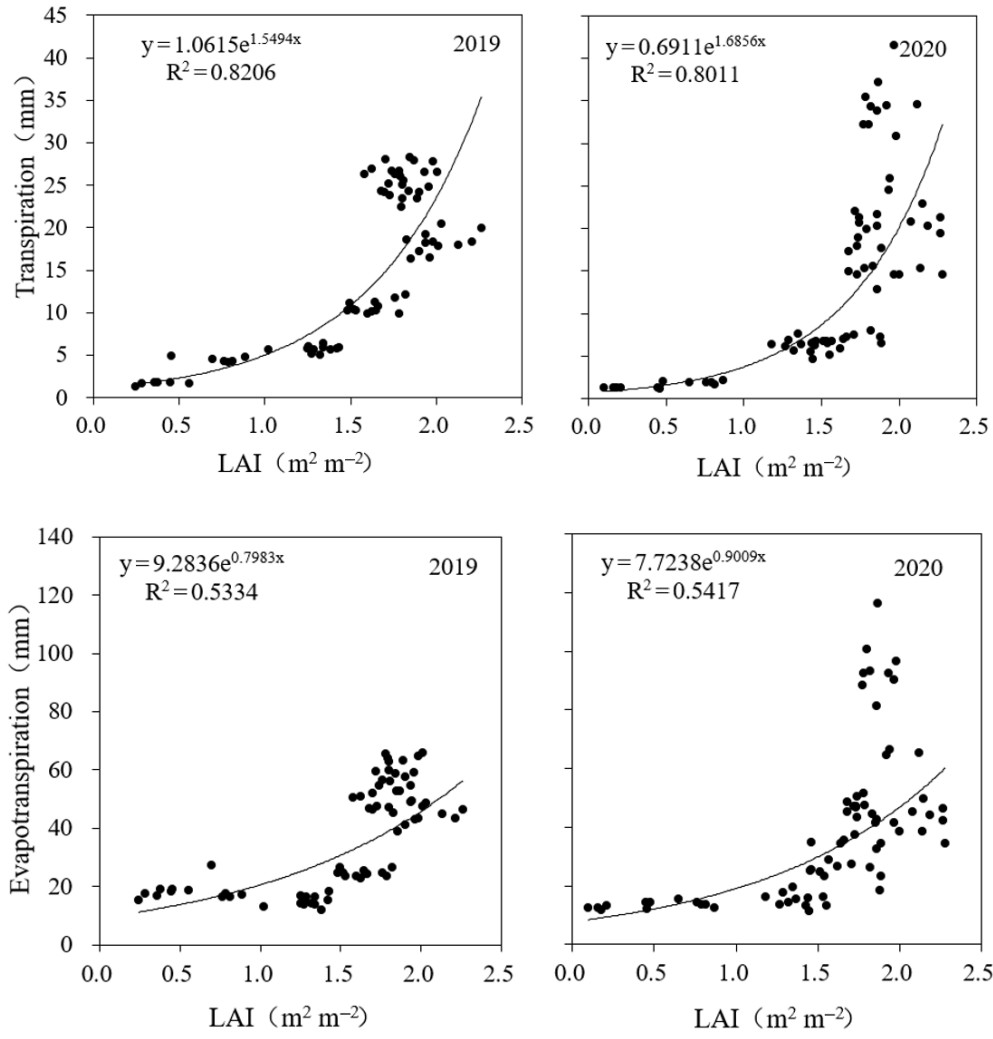

**Figure 5.** Relationship between LAI and transpiration (T) or evapotranspiration (ET) in 2019 and 2020.

### 3.5. Leaf Area Index and Crop Coefficient Relationship

Rainfalls, $ET_0$, ET under different water conditions, crop coefficient $Kc_I$ without water stress and water stress coefficient Ks and Kc under water deficit from April to September in 2019 and 2020 are shown in Table 4. As can be seen from Table 4, $ET_0 > ET_I > ET_D$ were shown in each year of the experiment, indicating that water conditions had a great influence on evapotranspiration, and irrigation increased the evapotranspiration of orchards in the study area.

The crop coefficient $Kc_I$ under no water stress changed with the development of the apple tree canopy. Because of canopy development, the wet evaporation surface area increases; thus, $ET_I$ also increases under the same atmosphere demand, resulting in an increase in the crop coefficient $Kc_I$. In 2019, the $K_{CI}$ rose from 0.43 in April to 0.74 in July, and decreased to 0.60 in September. In 2020, the $K_{CI}$ rose from 0.40 in April to 0.98 in August, and decreased to 0.68 in September. The reason for the decrease of KC after August may be related to the decrease of LAI. In both 2019 and 2020, the variation of Ks was basically consistent with the variation of rainfall, and its variation range is 0.53 to 1.15. From April to June, there was less rainfall and Ks was low, and apple trees suffered serious water stress. After July, the rainfall increased sharply and Ks was approximately 1.0, indicating that the growth and development of apple trees would not be affected by water stress. The variation trend of crop coefficient Kc under water deficit was basically the same as that of $Kc_I$ and Ks, which were small at the early growth stage and were 0.28 and 0.23, respectively. With the growth of apple trees, the crop coefficient Kc increased continuously, and reached the maximum value in August of 2019 and 2020, which was 0.81 and 0.91, respectively, and decreased to 0.64 and 0.67 in September.

**Table 4.** Variation of crop coefficient Kc under different soil moisture.

| Year | Month | Rainfall (mm) | $ET_D$ (mm) | $ET_I$ (mm) | $ET_0$ (mm) | $Kc_I$ | Ks | Kc |
|------|-------|---------------|-------------|-------------|-------------|--------|------|------|
| | 4 | 65.5 | 35.6 | 54.0 | 125.5 | 0.43 | 0.66 | 0.28 |
| | 5 | 8.9 | 29.3 | 55.4 | 131.9 | 0.42 | 0.53 | 0.22 |
| | 6 | 42.4 | 47.7 | 66.3 | 133.8 | 0.50 | 0.72 | 0.36 |
| 2019 | 7 | 90.5 | 100.8 | 109.6 | 148.1 | 0.74 | 0.92 | 0.68 |
| | 8 | 101.1 | 122.3 | 106.3 | 151.9 | 0.70 | 1.15 | 0.81 |
| | 9 | 91.1 | 89.1 | 83.2 | 138.7 | 0.60 | 1.07 | 0.64 |
| | Total | 399.5 | 424.8 | 474.8 | 829.9 | / | / | / |
| | 4 | 16.3 | 26.7 | 46.0 | 114.2 | 0.40 | 0.58 | 0.23 |
| | 5 | 10.7 | 29.0 | 51.7 | 125.0 | 0.41 | 0.56 | 0.23 |
| | 6 | 14.2 | 52.5 | 79.5 | 135.6 | 0.59 | 0.66 | 0.39 |
| 2020 | 7 | 58.3 | 95.2 | 117.6 | 148.8 | 0.79 | 0.81 | 0.64 |
| | 8 | 245.2 | 175.6 | 188.8 | 192.7 | 0.98 | 0.93 | 0.91 |
| | 9 | 67.2 | 84.2 | 85.9 | 126.3 | 0.68 | 0.98 | 0.67 |
| | Total | 411.9 | 463.2 | 569.6 | 842.6 | / | / | / |

Note: $ET_D$ and $ET_I$ represent evapotranspiration under water stress and irrigation, respectively.

LAI can reflect the growth status of crops, so the relationship between LAI and crop coefficient Kc can realize the dynamic simulation of evapotranspiration under the condition of water deficit. There is a significant exponential relationship between LAI and crop coefficient Kc (Figure 6), and the fitting equation was as follows: $Kc = 0.1141e^{1.0665LAI}$, $R^2 = 0.7055$, $p < 0.01$. These results indicated that LAI could be used to estimate the crop coefficient Kc of apple orchards under the condition of water deficit in the Loess Plateau, and the actual evapotranspiration of apple orchards in this region could be predicted.

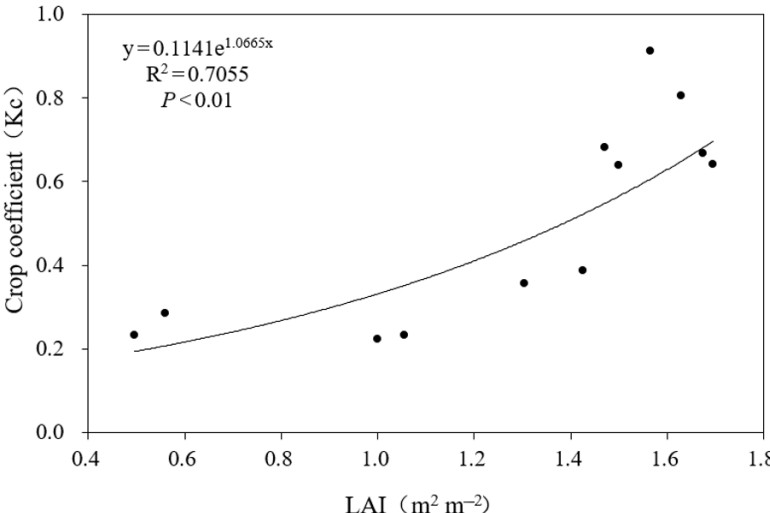

**Figure 6.** The relationship between leaf area index of apple trees and crop coefficient Kc during the growth period in 2019 and 2020.

## 4. Discussion

### 4.1. Leaf Area Index

LAI is an important parameter that characterizes the interface between a vegetation canopy and the atmosphere. It has been commonly used in plant growth, climatic and canopy reflectance models, as well as in much other research [60]. In the Loess Plateau region of China, the LAI of apple orchards is closely related to topography, soil, meteorological factors (such as rainfall and rainfall distribution, illumination, temperature, etc.) and management measures (such as density, canopy structure, soil moisture conservation measures, irrigation, fertilization and pest control). In this study, the variation trend of LAI in experiment orchards during the whole growing season was consistent with other results from previous studies [47,61–63]; LAI was smaller than the observation of others [47,61–63], which is determined by the growth of the apple tree under the local natural ecological conditions. The rainfall during the apple tree growing season was 399.5 mm and 411.9 mm in 2019 and 2020, respectively, the rainfalls were normal in experiment orchards and can represent the general rainfall conditions in the region. During the survey, apple tree pruning, fertilization, flower management, pest control, weeding and so on were normal. Therefore, we believe that the size and variation trend of the leaf area index investigated by us can represent the leaf growth status of local apple trees.

### 4.2. Precipitation and Reference Evapotranspiration

In the growth period of apples in 2019 and 2020, the rainfall distribution was not uniform. Most of the rainfall was concentrated in July to September, with the minimum rainfall in May and the maximum rainfall in July to August. In both 2019 and 2020, the evapotranspiration during the apple growth period was greater than the annual rainfall, which indicated that the rainfall during the growth period could not meet the demand of the water consumption of the apple orchard. The daily $ET_0$ of the apple orchard in 2019 and 2020 ranges from 1.98–6.88 mm $d^{-1}$ and 1.68–7.76 mm $d^{-1}$, respectively, and the mean values were 4.48 mm $d^{-1}$ and 4.52 mm $d^{-1}$, respectively. As can be seen from Table 2, rainfall and $ET_0$ are not closely related. The rainfall in April 2019 was 65.5 mm, 39.2 mm more than that in April 2020, but $ET_0$ was only 10.8 mm more. One reason may be that the low humidity soil absorbed and stored part of the rainfall, and the other reason was basically no grass covered on the ground at this stage, which affected the accuracy of $ET_0$ simulation by the PM equation [13], Alexandris et al. [14] pointed out that the FAO56-PM method, when applied above bare, non-irrigated soil, present overestimation. In addition, this period was the germination and leaf spreading period of apple trees, and the water consumption of transpiration was small, indicating that the large $ET_0$ was

mainly caused by soil evaporation. Rainfall is low in May and June, and the low soil moisture (or completely dry soil surface) and the persistence of dry conditions in the overlying to the soil surface air layer, results in energy allocation from the ground heat flux, attributing to the overlying to the soil's surface air layer. But grass had already covered the ground, so the estimated values of $ET_0$ in the experimental orchard can be considered as very close to $ET_0$ (grass reference height 8–12 cm). Strong winds are frequent in the experimental orchard, so the possible reason for the high $ET_0$ is wind speed. There was more rainfall and higher temperatures in July and August, the ground was full of weeds in the experimental orchard, $ET_0$ was high due to high transpiration consumption and high soil surface evaporation. After September, the temperature drops, the sunshine hours are shortened, and $ET_0$ becomes smaller.

The variation of daily and monthly $ET_0$ for the two years had no obvious regularity, partly because $ET_0$ is the result of a combination of meteorological factors, and the sensitivity of $ET_0$ to climatic factors varies on daily and seasonal scales [57]. Annual $ET_0$ was higher for 2019–2020 than that reported by Wang et al. [57] for the Loess Plateau. We speculate that the higher $ET_0$ was mostly caused by higher wind speed, low-relative humidity and more hours of sunshine. This was not in agreement with the views of Li et al. [64] and Wang et al. [56]. It should be noted here that the main influencing factors of $ET_0$ on the Loess Plateau vary with time and the regional climate environment. In addition, the PM equation was sensitive to errors induced by non-representative weather data; experimental weather stations situated above a dry soil will measure higher temperature and correspondingly, lower relative humidity values, compared to a station on well-irrigated soil [65,66]. This may result in an overestimation of $ET_0$.

### 4.3. Evapotranspiration

Evapotranspiration (ET) consists of soil evaporation (E), transpiration (T) and canopy interception (Ic). Soil evaporation could not be precisely measured on rainy days, so we present the results only for monthly ET. Quantification of the components of evapotranspiration is the key to designing management strategies for improving water use efficiency [47]. The factors that influence E, T and Ic would thus influence ET. In this study, E and T were the largest proportions of ET. E/ET was greater than T/ET, which is consistent with the research results of Li et al. [67] and Wang et al. [57]. The main reason is that the LAI was low in the orchard of our study, and the surface soil water after rainfall was the main source of ET loss [68]. However, both Wang [69] and Yu [70] believed that the largest component of ET was T, which may be related to tree age, density, rainfall and other natural ecological conditions. Ic was always small in the whole growth period, and Ic only accounted for 1.7–13.6% of ET. Ic showed different characteristics in different months, which was consistent with the research results of Li et al. [71]; however, Ic should not be ignored [61], because the physiological and ecological characteristics of plants (leaf area index, plant height) and various meteorological factors (rainfall, rainfall intensity and wind speed) will affect the plant canopy interception. If an evapotranspiration assessment does not consider canopy interception volume in apple orchards, it may be misleading regarding the relationship between water supply and demand. During the two years of the experiment, the maximum monthly ET in the apple orchard occurred in August. One reason is that this stage is during the rapid expansion of fruit and fruit development consumed a lot of soil water; another reason is more evaporation due to high temperatures and more rainfall. Because the apple trees in August were in the period of fruit expansion, during the fast growth rate, the fruit consumed a lot of water [46]. The minimum monthly ET appeared in May and April, This was because transpiration consumption was lower due to smaller leaf area and LAI, although less rainfall and strong winds caused more evaporation.

In this study, T and E were measured separately by thermal-dissipation probes and micro-lysimeters at localized scale. Each method for estimating T or E of whole apple orchards faced its own problems due to sampling constraints [57,72]. The limitation

of estimation of actual evapotranspiration in apple orchards was mainly related to the monitoring of soil evaporation by microlysimeters. Firstly, replacing the soil in the micro-lysimeters would be essential every 3–5 days and after a heavy rain (above 5 mm d$^{-1}$), which can become time consuming. Secondly, the time-resolution of the method was constrained by the weighing frequency of the micro-lysimeters [57].

### 4.4. Leaf Area Index and Evapotranspiration Relationship

Evapotranspiration intensity is jointly affected by soil moisture, solar radiation and crop LAI [73]. Our results showed that there was a positive correlation between LAI and evapotranspiration in apple orchards in the Loess hilly region, compared with the relationship between LAI and evapotranspiration, LAI is more closely related to transpiration (Figure 5), however, there was no significant correlation between LAI and soil evaporation. Our results were consistent with most previous research results. For example, Du et al. [46] reported that there was a positive linear relationship between apple transpiration and LAI at the stages of leaf expansion and fruit expanding in apple orchards in Northwest China under irrigation conditions, however, there was no correlation between LAI and transpiration at the stages of bud development and flowering, and fruit maturing because the leaves were not developed during the stage of bud development and flowering, and the leaves withered during the stage of fruit maturing. Liu et al. [63] reported that there was a positive linear correlation between LAI and sapflow in apple orchards in arid areas of Northwest China. Testi et al. [29] reported that there was a significant linear growth relationship between evapotranspiration and LAI of olive trees in southern Spain when there was no moisture restriction in summer. Wang et al. [74] reported evapotranspiration, canopy interception, and transpiration were all significantly positively correlated with the canopy LAI of the plantation, while soil evaporation was significantly negatively correlated with canopy LAI. Juhász and Hrotkó [75] reported that canopy transpiration was positively correlated with LAI of sweet cherry in Hungarian. Benyon et al. [76] found that there was a positive correlation between annual transpiration and LAI of Radiata Pine and Eucalyptus globulus in south-eastern Australia. Almeida et al. [77] reported that forest evapotranspiration was positively correlated with LAI of Eucalyptus grandis on the Atlantic coast of Brazil. Tian et al. [78] studied the response of water yield to key stand structure and site factors in the Liupanshan larch plantation, established and fitted canopy LAI and stand evapotranspiration models in the growing season, and calculated water yield in the growing season using water budget method. Zhang et al. [79] reported that LAI of winter wheat was positively correlated with evapotranspiration in Changwu tableland, Shaanxi.

However, there are a few studies that differ from our results. Juul et al. [80] found that there was a significant negative correlation between evapotranspiration and LAI of Betula latticola in the Netherlands. Li et al. [81] reported evapotranspiration and transpiration showed a curve growth with LAI of apple trees under sufficient water supply.

In this study, the rainfall during the apple tree growing season was 399.5 mm and 411.9 mm in 2019 and 2020, respectively and the rainfalls were the general rainfall conditions in the region. Rainfall varies widely from year to year, usually by approximately 20% from year to year. This difference in rainfall will inevitably lead to differences in the growth of apple branches and leaves. Therefore, the relationship between LAI and ET in rain-deficient years and rain-rich years needs to be further studied.

### 4.5. Leaf Area Index and Crop Coefficient Relationship

Crop coefficient Kc reflects the comprehensive influence of crop growth, biological characteristics, yield level and other conditions on water demand. The crop coefficient approach is based on the assumption that the reference evapotranspiration $ET_0$ accounts for the atmospheric effect on the evapotranspiration process, whereas the crop coefficient Kc describes the integrated effects of aerodynamic and canopy resistances to water vapor flux, and soil evaporation [2]. Crop water demand can be calculated according to $ET_0$ and

crop coefficient Kc [42]. In this study, the crop coefficient Kc was defined as the product of the crop coefficient $Kc_I$ under no water stress and the water stress coefficient Ks under water stress, namely $Kc = Kc_I \times Ks$.

The crop coefficient $Kc_I$ under no water stress changes with the development of the apple tree canopy. Because of canopy development, the wet evaporation surface area increases; thus, ET also increases under the same atmosphere demand, resulting in an increase in the crop coefficient $Kc_I$. The variation trend of $Kc_I$ during the whole growing season was consistent with LAI, and it agreed with other results from previous studies [42,47]. There was an excellent linear relationship between crop coefficient $Kc_I$ and LAI, indicating that crop coefficient $Kc_I$ can be estimated using LAI. In this study, the crop coefficient $Kc_I$ under no water stress was small, as compared to the results of Gong et al. [47]. LAI is also smaller than the results of Gong et al. [47], indicating the smaller $Kc_I$ may be related to smaller LAI.

If there is water stress under natural conditions, it is necessary to consider the water stress coefficient Ks to estimate the actual crop evapotranspiration. In this study, the variation of Ks is basically consistent with the variation of rainfall. From April to June, there was less rainfall and Ks was low, and apple trees suffered serious water stress. After July, the rainfall increased sharply and Ks was approximately 1.0, indicating that the growth and development of apple trees would not be affected by water stress. The reason of Ks from April to June is that a low soil water availability increases the difficulty of root uptake water, which stimulates the production of abscisic acid (ABA) in roots. An increase in root-source ABA content in leaves should decrease the opening of leaf stomata, which limits the transpiration of plants [47]. Our results showed that there was a significant exponential relationship between LAI with crop coefficient Kc ($p < 0.01$) under water deficit. It indicated that the index model established by LAI can accurately estimate the crop coefficient Kc, which further provides a theoretical basis for the prediction of evapotranspiration under the condition of water deficit in Chinese Loess Plateau.

It should be noted that this study was carried out under specific site conditions. However, in the Loess Plateau region of China, the site conditions of apple orchards, such as elevation, topography, slope and slope position vary greatly, which inevitably leads to great differences in meteorological data. Therefore, the relationship between LAI and KC under different site conditions needs to be further studied.

## 5. Conclusions

The change of LAI in apple orchards showed a rapid increase—moderate increase—declined trend from April to September in the Loess hilly region of northern Shaanxi in China. The variation range of LAI was 0.26–2.16 [$m^2 \ m^{-2}$].

The ET of the orchard was greater than the rainfall; the maximum ET was in July or August. The maximum components of ET in apple orchards was E, and E accounted for 47.8–49.1% of ET; T accounted for 42.5–43.9% of ET and Ic accounted for only 9.1–9.6% of ET. There was a significant exponential relationship between LAI and T or ET. The crop coefficient $Kc_I$ under no water stress changed with the development of the apple tree canopy. The variation of water stress Ks was basically consistent with the variation of rainfall. There is a significant exponential relationship between LAI and crop coefficient Kc under water deficit ($Kc = 0.1141e^{1.0665LAI}$, $R^2 = 0.7055$, $p < 0.01$). LAI could be used to estimate the crop coefficient Kc of apple orchards under water deficit in the Loess Plateau, and the actual evapotranspiration of apple orchards in this region could be predicted. This study was carried out under normal rainfall conditions and specific site conditions. Therefore, the relationship between LAI and KC should be further studied under the conditions of low rainfall years and rich rainfall years, as well as different site conditions.

**Author Contributions:** Conceptualization, Q.J. methodology, Q.J. and Y.-P.W.; formal analysis, Q.J.; data curation, Q.J.; writing-original draft preparation, Q.J.; writing-review and editing, Q.J. and Y.-P.W.; supervision, Y.-P.W.; project administration, Y.-P.W.; funding acquisition, Y.-P.W. Both authors have read and agreed to the published version of the manuscript.

**Funding:** Contract/grant sponsor: National Natural Science Foundation of China; contract/grant number: 41571218, 41401613. Contract/grant sponsor: Science-Technology Innovation Project of Shaanxi; contract/grant number: 2021NY-204.

**Institutional Review Board Statement:** Not applicable.

**Informed Consent Statement:** Not applicable.

**Data Availability Statement:** Data sharing not applicable.

**Acknowledgments:** We are grateful for the support from the staff of the Mizhi Experimental Station of Northwest A&F University.

**Conflicts of Interest:** The authors declare no conflict of interest.

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
