# Peer review of "Relationships between Leaf Area Index and Evapotranspiration and Crop Coefficient of Hilly Apple Orchard in the Loess Plateau"

_water, doi:10.3390/w13141957_

Round 1

Reviewer 1 Report

The authors have submitted a reliable experimental paper. The document has been carefully prepared, and all the sections are presented in the right sequence. In addition, the authors described the study area, the equipment, and the experimental process in detail and clearly.

However, there are some vague points (serious) in the manuscript on the fundamental concept of reference evapotranspiration. For example, the ETo estimations from equation (6) can probably give deviations in final results in equations (7 & 8).

There is no clarification in the manuscript on how the ETο is calculated in the study area. ETo requires the meteorological data to be taken above standardized well-watered and vegetation-covered surfaces (usually, grass reference surface). The "0.34" in equation (6) represents the wind coefficient for the reference crop (grass) in sec/m. Although the paper of Allen et al. [24] has more than 23688 citations in literature (Google Scholar citations), the number of scientists who point out the importance and significance of a standardized vegetation surface is quite limited. 

 On the other hand, reference [29] conflicts with the reference [24]. Even the reference title [29] conflicts with the concept of reference evapotranspiration and has a serious incompatibility. Unfortunately, there is much confusion on the concept (s) of evapotranspiration in the literature leading to misinformation like a domino. Therefore, suggest that the reference [29] be removed.

However, the submitted paper can be published if the authors justify the ΕΤο assessments and enrich the discussion accordingly. The following references could help the authors in this direction.

Alexandris, S., & Proutsos, N. (2020); Awal et al. 2020); Eching et al. (2002); Yoder et al. 2000; Temesgen et al. (1999)

Also, Line 21: The physical meaning of “LAI” defines the plan area of leaves per unit ground area, but the unit is a dimensionless number and denotes as “[ ].” (Smith, et al., 1994; Monteith, J., & Unsworth, M. (2013).

Reviewer 2 Report

Ms. Ref. No.: water-1282186

Title: Relationships between Leaf Area Index and Evapotranspiration and Crop Coefficient of Hilly Apple Orchard in the Loess Plateau

Journal: Water

These results suggest are important and its representation will improve the vegetation properties in LSMs is necessary for analyzing land–atmosphere interactions may be a function of vegetation properties such as leaf area index (LAI). However, there are some points described below that have to be considered before publication. For instance, though authors have mentioned the literature survey part, it fails to provide clear view on the previous attempts for understanding the relationship between Kc and LAI. Estimating the impact of such changes on a wide range of ecosystem services is seldom attempted. However, there are some points described below that have to be considered before publication. The overall presentation in the Introduction section lacks synergy and exists in bits and pieces. Though authors have identified the research gaps the literature survey part can be more streamlined and while coming towards the problem statement. The introduction section need some rework and restructuring to make a precise outline of study that can be achieve by making discussion in following sequence,

  1. Need - challenges - methods of using LAI and Kc with different satellite products and hydrological models
  2. Identified research gaps or problem statement
  3. Proposed solution to address the problem statement or to fill the research gaps
  4. Objectives in line of identified research gaps or problem statement

-There is a need to clearly state the objective(s) of the study towards the end of the introduction.

- Discussion and Conclusion sections can be more rigorous with objective base.

Currently, many of the statements are not supported by published works. Authors may like to find studies in line of their statements to add the scientific weight in their observations. For instance, at line 56-66 and 68-70: authors can include some of the recent studies to discuss the application of genetic algorithm in the crop models. Some of the good reviews are given below which they can include to make their introduction better. There is a lot of research out there on these topics.  

The authors must review the recent literature to provide a clearer context for this work.  For this I would like to suggest authors to consider adding some of these recent literature that will be useful for the following manuscript in attributing/interpreting some of your statements in the introduction section. 

Srivastava, A., Sahoo, B., Raghuwanshi, N. S., & Singh, R. (2017). Evaluation of variable-infiltration capacity model and MODIS-terra satellite-derived grid-scale evapotranspiration estimates in a River Basin with Tropical Monsoon-Type climatology. Journal of Irrigation and Drainage Engineering, 143(8), 04017028. https://doi.org/10.1061/(ASCE)IR.1943-4774.0001199

The authors mention several studies on using the use of evapotranspiration using crop coefficient based approach. However, the findings or conclusions in terms of their modelling approaches are not mentioned. Elaborate more on how this paper differs or affirms the findings and conclusion of those studies. These points need to be clearly addressed in the introduction section.

57-60 I have a big and key concern for these lines. Authors have missed discussing the important aspect of incorporation of the effect of vegetation cover on water cycle. There is a vast literature on this I would like to suggest few lines following to this which author should add is “Vegetation has major impact on the different hydrological components of the water cycle which alters the surface energy balance. Heterogeneity in vegetation affects the physical characteristics such as albedo, atmospheric transmissivity, root zone soil moisture, crop growth, evapotranspiration, leaf area index (LAI), stomatal conductance, and surface runoff (Srivastava et al., 2020; Aghsaei et al., 2020)”.  I would recommend adding these recent references to add more scientific weight in their Introduction.

Srivastava, A., Kumari, N. & Maza, M. (2020). Hydrological Response to Agricultural Land Use Heterogeneity Using Variable Infiltration Capacity Model. Water Resour Manage 34, 3779–3794. https://doi.org/10.1007/s11269-020-02630-4

Aghsaei, H., Dinan, N. M., Moridi, A., Asadolahi, Z., Delavar, M., Fohrer, N., & Wagner, P. D. (2020). Effects of dynamic land use/land cover change on water resources and sediment yield in the Anzali wetland catchment, Gilan, Iran. Science of the Total Environment, 712, 136449.

I suggest to author to give a table giving the information of datasets used in this study with their source and time period. Also provide a time series of analysis of the rainfall in the study region with the variation of evapotranspiration.

Authors should mention in the conclusion where these models/relationship applied need to be improved or any suggestions for future research in terms of models' performance improvement and application

Also, it would be interesting to see if authors can use remote sensing datasets and provide the verification of satellite data how it is predicting the relationship of LAI and Kc.

Round 2

Reviewer 2 Report

The authors have addressed all previous concerns expressed by the reviewers and in the process have improved the work, confirmed the validity of their findings and gained confidence in their introduction, methods, results and conclusions. I would like to congratulate the authors for an interesting and well executed work and I recommend this manuscript for publication in Water (MDPI) in its current form.

This manuscript is a resubmission of an earlier submission. The following is a list of the peer review reports and author responses from that submission.